# Dipeptidyl Peptidase-4 Inhibitor-Related Bullous Pemphigoid: Clinical, Laboratory, and Histological Features, and Possible Pathogenesis

**DOI:** 10.3390/ijms232214101

**Published:** 2022-11-15

**Authors:** Chih-Tsung Hung, Yung-Lung Chang, Wei-Ming Wang

**Affiliations:** 1Department of Dermatology, Tri-Service General Hospital, National Defense Medical Center, Taipei 114, Taiwan; 2Graduate Institute of Medical Sciences, National Defense Medical Center, Taipei 114, Taiwan; 3Department of Biochemistry, National Defense Medical Center, Taipei 114, Taiwan

**Keywords:** dipeptidyl peptidase-4 inhibitor, vildagliptin, keratinocyte, bullous pemphigoid, IL-6, dermoepidermal junction

## Abstract

Dipeptidyl peptidase-4 inhibitor (DPP4i) is a widely used antidiabetic agent. Emerging cases of DPP4i-associated bullous pemphigoid (DBP), whose pathogenesis remains unclear, have been reported. Thus, a retrospective study was conducted from January 2016 to June 2021 to determine the clinical, laboratory, and histopathological features of DBP and idiopathic bullous pemphigoid (IBP). We set up in vitro experiments using vildagliptin-treated HaCaT keratinocytes to validate what we found by analyzing published RNA sequencing data about the genes related to the dermal–epidermal junction. We also observed IL-6 expression by HaCaT cells treated with vildagliptin. We enrolled 20 patients with DBP and 40 patients with IBP. The total Bullous Pemphigoid Disease Area Index (BPDAI) score was similar in both groups. However, the BPDAI score of erosions and blisters in DBP was significantly higher than that in IBP (24.6 vs. 16.68, *p* = 0.0189), and the score for urticaria and erythema was lower in DBP (12 vs. 19.05, *p* = 0.0183). The pathological features showed that the mean infiltrating eosinophil number per high-power field was significantly lower in DBP than in IBP (16.7 vs. 27.08, *p* = 0.023). The expression of *LAMA3*, *LAMB3*, *LAMC2*, *DST*, and *COL17A1* decreased significantly in vildagliptin-treated human keratinocytes. On the other hand, IL-6, the hallmark cytokine of bullous pemphigoid (BP) severity, was found to be upregulated in HaCaT cells by vildagliptin. These experimental findings imply less of a requirement for eosinophil infiltration to drive the inflammatory cascades in DBP blistering. Both immunologic and non-immunologic pathways could be employed for the development of DBP. Our findings may help explain the higher incidence of non-inflammatory BP that was observed in DBP.

## 1. Introduction

Bullous pemphigoid (BP), the most common autoimmune bullous disease, usually affects the elderly [1]. Several triggering factors, such as drugs, vaccination, radiotherapy, and ultraviolet radiation, have been reported to induce BP [2,3,4]. The pathogenesis of drug-induced BP is uncertain, and various medications have been reported to be associated with BP [5,6]. Commonly reported medications included neuroleptics, aldosterone antagonists, PD-1/PD-L1 inhibitors, and recently dipeptidyl peptidase-4 inhibitor (DPP4i) [6,7,8,9].

DPP4i is an oral anti-hyperglycemic agent for type 2 diabetes mellitus (DM) [10]. DPP4i blocks the degradation of the incretin hormones, such as glucagon-like peptide-1 (GLP-1) and glucose-dependent insulinotropic polypeptide, and enhances glucose-dependent insulin secretion and the reduction in postprandial glucagon [11]. In recent years, several studies have shown that DM patients taking DPP4i are more predisposed to BP [12,13,14,15,16].

Our previous study revealed that patients taking DDP4is had a higher risk of BP, with an adjusted hazard ratio of 2.382, via the National Health Insurance Research Database of Taiwan [17]. The study also demonstrated that vildagliptin was associated with the highest risk of BP. Similar results were also observed in Japan’s nationwide retrospective observational study [243 with DBP and 461 with IBP, vildagliptin (37.2%)] and China’s retrospective analysis of databases [114 with DBP, vildagliptin (52.63%)] [18,19]. These findings remind clinicians to pay more attention when evaluating patients with DM and prescribing DPP4i for them.

According to previous studies, the clinical presentation of DPP4i-related BP (DBP) seems different from that of idiopathic BP (IBP). A retrospective study showed that the lesions of DBP were frequently non-inflammatory or pauci-inflammatory [20]. Chijiwa C et al. [21] reported that significantly fewer eosinophils were infiltrating the lesions of DBP than of IBP. In addition, a lower positive rate of the BP180 NC16A antibody was observed in patients with DBP than in those with IBP [14]. Izumi et al. [20] identified autoantibodies that targeted the mid-portion of Collagen 17 but not the NC16A region of BP180 in non-inflammatory BP cases. Controversially, a recently published paper reported higher levels of eosinophil-attractive cytokines in DBP skin samples, which correlated with serum anti-BP180 NC16A IgG autoantibody titers [22]. These emerging study results make the underlying pathogenic mechanisms of DBP more intriguing; however, they remain unclear [23].

In light of the aforementioned publications, we conducted a retrospective study to determine the clinical, laboratory, and histopathological features of DBP and IBP cases collected in our institution. Furthermore, we also set up experiments to explore how DPP4i may affect human keratinocytes in the gene expression of relative factors for BP development, including cytokines and adhesion molecules distributed in the dermoepidermal junction where blistering in BP occurred. We attempted to find out the possible mechanisms for blister formation in DBP in the presence of a few eosinophils.

## 2. Results

### 2.1. Characteristics of Patients with DBP and IBP

We enrolled 60 patients with BP, including 20 patients with DBP and 40 patients with IBP, in this retrospective study. The age and sex distributions were similar in the two groups (Table 1). Regarding the clinical presentations (such as pruritus and mucosal involvement), there was no significant difference between the two groups.

The total BPDAI score in DBP was higher than that in IBP (37.1 vs. 36.33); however, the difference was not statistically significant (*p* = 0.8927). In the subgroup analysis of the components of the BPDAI, the score for erosions and blisters in DBP was significantly higher than that in IBP (24.6 [3–65] vs. 16.68 [3–42], *p* = 0.0189). On the other hand, the scores for urticaria and erythema were lower in DBP than in IBP (12 [0–40] vs. 19.05 [2–46], *p* = 0.0183). There was no significant difference in mucosal lesions between DBP and IBP (0.5 [0–5] vs. 0.6 [0–7], *p* = 0.836).

Regarding hematological involvements, there was no significant difference in anemia and thrombocytopenia between the two groups. The absolute eosinophil count was higher in DBP than in IBP (780 [0–3138] vs. 470 [0–1672]); however, the difference was not statistically significant (*p* = 0.4216). The median latency between the use of DPP4i and the diagnosis of BP was 14.5 (1–30) months.

### 2.2. Pathological Assessment of Patients with DBP and IBP

The number of infiltrating eosinophils per high-power field was lower in DBP than in IBP (16.7 [0–45] vs. 27.1 [0–65], *p* = 0.023; Figure 1). Regarding the DIF results, linear IgG deposition along the dermoepidermal junction was higher in DBP than in IBP (85% vs. 72.5%, *p* = 0.2884), and linear C3 deposition was also higher in DBP than in IBP (90% vs. 72.5%, *p* = 0.125); however, neither of the differences attained statistical significance (90.0% vs. 83.8%, *p* = 0.395).

### 2.3. Heatmap of Expressed Genes Related to the Dermal–Epidermal Junction in DPP4i-Treated vs. Vehicle-Treated Primary Keratinocytes

Considering the lower number of eosinophils infiltrating the skin of patients in DBP than in IBP, some other non-inflammatory pathways could be involved in the pathogenesis of the disease. We analyzed the published RNA sequencing data of primary adult human keratinocytes treated with DPP4i [24]. We checked the gene-encoded proteins related to the dermal–epidermal junction, including *LAMA3*, *LAMB3*, *LAMC2*, *LAMB1*, *LAMC1*, *ITGA6*, *ITGB4*, *ITGA3*, *ITGB1*, *DST* (encoding dystonin, known as BPAG1 or BP230), *COL17A1* (encoding the alpha chain of type 17 collagen, known as BPAG2 or BP180), *PLEC*, and *HSPG2* (Figure 2). The results revealed that the expression of *LAMA3, LAMB3, LAMC2, DST,* and *COL17A1* decreased significantly.

### 2.4. Vildagliptin Decreased the Expression of DYS and Collagen 17A1 in a Time-Dependent Manner

Following the above heatmap analysis, we were interested in the expression of both well-known BP antigens from keratinocytes in the absence or presence of vildagliptin, since this medication has been shown to be the most common culprit of DBP [17,18,19]. We used vildagliptin (5 µM) to treat HaCaT cells with increasing incubated durations and then detected the mRNA levels of respective genes. Interestingly, the results revealed that the expression of DYS and Collagen 17A1 decreased in a time-dependent manner (Figure 3).

### 2.5. Vildagliptin Decreased the Expression of LAMA3, LAMB3, and LAMC2 in a Time-Dependent Manner

Next, we validated *LAMA3*, *LAMB3*, and *LAMC2* genes via qPCR according to the same experimental setting for drug incubation and using respective primers to detect the indicated target genes. The results revealed that the expression of *LAMA3*, *LAMB3*, and *LAMC2* indeed decreased with vildagliptin treatment in a time-dependent manner (Figure 4).

### 2.6. The Expression of IL-6 Was Stimulated by Vildagliptin-Treated HaCaT Cells

IL-6 has been considered a biomarker of BP disease activity and severity [25]. It has been demonstrated to be secreted by different cells, including keratinocytes and eosinophils. Surprisingly, the expression of IL-6 was stimulated by vildagliptin in HaCaT cells (Figure 5a). IL-6 has been said to have a putative truncation site for DPP4, indicating that secreted IL-6 may accumulate in the presence of DPP4i [26]. Importantly, the endogenous IL-6 expression level increased in a time-dependent manner when the HaCaT cells were incubated with exogenous IL-6 (Figure 5b). The data shown in Figure 5 suggest the allowance of less of a requirement of eosinophil infiltration to drive the inflammatory cascades in the pathogenesis of DBP blistering. DPP4i-treated keratinocytes may play a role in providing sufficient IL-6 in the skin of DBP patients despite the decreased presence of inflammatory eosinophils.

## 3. Discussion

According to our findings, patients with DBP had several distinctive clinical, immunological, laboratory, and histological characteristics. The median latency period between the use of DPP4i and the onset of BP was 14.5 months. In line with previous reports, there was a long latency period from drug prescription to the onset of BP [12,13,27]. Rather than simply being an adverse drug reaction, DBP should be considered a drug-aggravated disease [27,28].

In terms of disease severity, the overall BPDAI score in DBP was similar to that in IBP in our study, which is consistent with the results of large studies conducted in Japan [18], France [29], and Thailand [27]. However, in the subgroup analysis of BPDAI components, patients with DBP had more severe erosions and blisters but milder urticarial and erythematous lesions than those with IBP. These clinical features can be denoted as non-inflammatory BP. There have been several studies that showed comparable clinical findings (less erythema and fewer urticarial lesions) in patients with DBP. First, Izumi et al. [20] reported a non-inflammatory form in seven patients with DBP characterized by scant erythema and sparse eosinophilic infiltration. Subsequently, several studies reported the similar finding that the non-inflammatory phenotype is more frequently observed in DBP [27,30,31,32,33]. However, several European studies did not reveal unique clinical and immunological characteristics among patients with DBP and those with IBP [29,34,35]. This finding implies that ethnic discrepancy plays a role in the phenotype of DBP.

Our histopathological findings demonstrated that infiltrating eosinophils in the dermis were significantly fewer in patients with DBP than in those with IBP. Several studies also reported a similar trend with statistical significance [21,30,36]. Therefore, it is reasonable to hypothesize that there is another mechanism other than the inflammatory pathway in the pathogenesis of DBP, and it may correlate with the clinical presentation of a non-inflammatory variant of BP.

The pathogenesis of DBP remains largely unknown, and several important findings were observed. DPP-4, previously known as the T cell surface marker cluster of differentiation 26 (CD26), is a transmembrane glycoprotein expressed in various kinds of cells, including keratinocytes, endothelial cells, and immune cells [37,38]. DPP4 expressed on T cells modulates several functions, including T cell proliferation, migration, inflammation, and auto-activation [39]. DPP4 works as an important co-stimulatory signal for CD4^+^T cell activation and regulates lymphocyte–epithelial cell adhesion [40]. A previous study showed that regulatory T (Treg) cells with suppressive abilities do not proliferate in response to autoreactive T cells in patients with DBP [41]. DPP4 also functions as a cell surface plasminogen receptor, which enables the conversion of plasminogen into plasmin [42], a serine protease that cleaves BP180 within the NC16A domain [43]. Therefore, the suppression of DPP4 may develop new epitopes for DBP autoantibodies and lead to the immunologic intolerance of BP180. Additionally, Izumi et al. [20] identified autoantibodies targeting the mid-portion of BP180 instead of the NC16A domain in non-inflammatory BP cases.

Considering the higher incidence of the non-inflammatory phenotype observed in DBP, we believed there would be mechanisms other than the inflammatory pathway in the pathogenesis of DBP. We analyzed the published RNA sequencing data of primary adult human keratinocytes treated with a DPP4i, 3-N-[(2S,3S)-2-amino-3-methylpentanoyl]-1,3-thiazolidine (BioVision, Inc., Milpitas, CA, USA) [24]. The results revealed the upregulation of the late cornified envelope cluster. Interestingly, as shown in Figure 2, gene-encoded adhesion proteins related to the dermal–epidermal junction were downregulated significantly, especially *LAMA3*, *LAMB3*, *LAMC2*, *DST*, and *COL17A1*. However, the potent and selective DPP4i, 3-N-[(2S,3S)-2-amino-3-methylpentanoyl]-1,3-thiazolidine, is not prescribed clinically for type 2 DM treatment. Our previous data showed that vildagliptin was the most common culprit agent [17], which is in line with the findings of other studies [12,13,33]. Therefore, we used vildagliptin (5 µM) to treat HaCaT cells and then checked the mRNA of *LAMA3*, *LAMB3*, *LAMC2*, *DST*, and *COL17A1* via qPCR. In support of our findings, the mRNA of these genes decreased significantly in a time-dependent manner. Mutations of *LAMA3*, *LAMB3*, *LAMC2*, *DST,* and *COL17A1* have been reported in heritable skin fragility disorders and epidermolysis bullosa (EB) [44,45]. The clinical presentation of EB is characterized by mucocutaneous fragility and blister formation easily induced by minimal trauma. The histologic presentation of DBP is comparable with the characteristic cell-sparse blisters of EB. We speculate that less inflammation was required to induce subepidermal blister formation in DBP due to the potential fragility of the basement membrane caused by the decreased expression of *LAMA3*, *LAMB3*, *LAMC2*, *DST*, and *COL17A1* from DPP4i-treated keratinocytes. Our findings may help explain the higher incidence of non-inflammatory BP that was observed in DBP.

Although we found that the decreased expression of genes related to the dermal–epidermal junction could be involved in the pathogenesis of DBP as a non-inflammatory pathway, we were also interested in the effects exhibited by IL-6, the well-known major inflammatory cytokine and biomarker of BP severity [25]. However, the time-dependent decreased expression of genes related to the dermal–epidermal junction was not observed in the IL-6 treatment of HaCaT cells (data not shown). This suggests that vildagliptin works differentially from IL-6 in the perturbation of the adhesion integrity of the dermal–epidermal junction, which causes blister formation in BP.

Dermal infiltration by eosinophils has been thought to play a predominant role in the pathogenesis of BP [46] and to represent the histologic hallmark of BP lesions [47]. Considering the puzzling phenomenon of less eosinophil infiltration shown in the skin lesions of DBP, we wondered what might be a reasonable explanation for the full-blown clinical manifestation of blistering. IL-6 has been considered pivotal in driving various autoimmune disorders [48,49]. Keratinocytes have also been demonstrated to release IL-6 under different stimulations and to participate in the pathogenesis of BP [47,50]. We revealed that the expression of IL-6 can be stimulated by vildagliptin in human keratinocytes (Figure 5a). Conceivably, the IL-6 released from keratinocytes may be accumulated in the skin tissue because IL-6 is a potential substrate subject to N-terminal cleavage by DPP4, whose enzymatic activity would be abrogated by DPP4i [26]. The surrounding increased IL-6 may feedback to stimulate more IL-6 production from keratinocytes and drive the downstream inflammatory cascades of DBP [47]. In support of this, a recently published article demonstrated comparably elevated *IL-6* mRNA and protein levels in both lesional and non-lesional skin tissues of either IBP or DBP patients [22]. Importantly, it has also been reported that BP disease activity can be controlled by the modulation of IL-6 production by keratinocytes [51]. Currently, we are working on an exploration of the underlying molecular mechanism employed by DPP4i in the regulation of *IL-6* expression. Our investigation will hopefully bring clarification and assist in developing therapeutic strategies in the future.

The main strength of our study is that we found a potential non-immunological pathway driven by DPP4i in the pathogenesis of DBP. Additionally, we found that DPP4i could also have immune-modulatory functions via the regulation of IL-6 production in DBP pathogenesis. DPP4i-treated keratinocytes may actively participate in the pathogenesis of DBP, since our data revealed a positive feedback loop for IL-6 over-expression by keratinocytes. However, our study also had some limitations. First, its retrospective design may inevitably mean incomplete data collection. The clinical and laboratory findings may have been recorded at different periods of diagnosis and follow-up. Second, this study was conducted in a tertiary referral center, which means there could be a selection bias of more severe and recalcitrant cases. Furthermore, the study was performed in a homogenous population of Taiwanese individuals; therefore, the results may not be universally applicable to all races. Our laboratory data might not answer the discrepancy of the clinical profile shown in different population-based studies. Third, DBP has a long latency period. DPP4i may exhibit various immune-modulatory functions in BP progression [52]. Although our in vitro laboratory data on IL-6 are in line with recent reports [22], the highly complicated in vivo pathogenic mechanisms involved are still largely unexplored. Finally, we only treated HaCaT cells with vildagliptin, the consistently reported most common culprit of DPP4i. We do not know whether the findings of this study truly reflect a mechanism generally employed by DPP4i or whether there are other unexplored intraclass differences, and this requires further investigation.

## 4. Materials and Methods

### 4.1. Patients

Patients diagnosed with BP between 1 January 2016 and 30 June 2021 at the Department of Dermatology, Tri-service General Hospital, National Defense Medical Center, Taiwan, were enrolled in this retrospective study. The diagnosis of BP was based on characteristic clinical features, typical histopathological findings, and at least one of the linear deposits of IgG and/or C3 along the basement membrane by direct immunofluorescence (DIF). Our study was approved by the Institutional Review Board of Tri-Service General Hospital (TSGHIRB No. A202105057).

### 4.2. Evaluation of Clinical Characteristics of BP

The clinical features were evaluated by two dermatologists based on patients’ clinical status or photo documentation. The BPDAI was calculated in 60 patients (20 DBP and 40 IBP) via a previously published method [53].

### 4.3. Laboratory Findings

Blood examinations, including hemoglobin levels and platelet counts, were assessed at the first presentation before any therapeutic procedures. Additionally, the absolute eosinophil count was classified into three severity groups [54]. Eosinophilia refers to an absolute eosinophil count of ≥500 cells/µL. The moderate-to-severe degree was defined by a cut-off value of ≥1500 cells/µL.

### 4.4. Histology and Immunopathology

Histopathological specimens from enrolled patients were all available. Specimens were reviewed by a blinded dermatopathologist, and the infiltrating eosinophils were counted under a high-power field. DIF was also evaluated for the presence of bound antibodies (IgG, IgM, and IgA) and complement (C3).

### 4.5. Cell Culture

Cells of a spontaneously immortalized keratinocyte cell line (HaCaT cells) were maintained in DMEM (10-013-CV, Corning™, Corning, NY, USA) supplemented with 10% fetal bovine serum, penicillin (10,000 IU/mL), and streptomycin (10,000 μg/mL) (30-002-CI, Corning™). The cells were treated with 5 μM vildagliptin (HY-14291, MedChemExpress, Monmouth Junction, NJ, USA) and harvested at different times for the time-course experiment.

### 4.6. Quantitative Real-Time PCR

Total cellular RNA was isolated using the TRIzol Reagent (15596018, Life Technologies). The extracted RNA was reverse-transcribed into first-strand cDNA from 1 µg of total RNA using the iScript™ cDNA Synthesis Kit (1708890, Bio-Rad, Hercules, CA, USA). Quantitative RT-PCR amplification of the target gene was performed using iQ™ SYBR^®^ Green Supermix (1708880, Bio-Rad). The PCR primers were as follows: *DST* sense 5′-GATGCAGATCCGAAAACCCCT-3′ and antisense 5′-CTCAGTGCGGTCCAGTTGTA-3′; *COL17A1* sense 5′-TTCAGAGGCATCGTTGGACC-3′ and antisense 5′-ATGACAAGCCGGCAGTATGT-3′; *LAMA3* sense 5′-CACCGGGATATTTCGGGAATC-3′ and antisense 5′-AGCTGTCGCAATCATCACATT-3′; *LAMB3* sense 5′-GCAGCCTCACAACTACTACAG-3′ and antisense 5′-CCAGGTCTTACCGAAGTCTGA-3′; and *LAMC2* sense 5′-GACAAACTGGTAATGGATTCCGC-3′ and antisense 5′-TTCTCTGTGCCGGTAAAAGCC -3′. The emission intensity from SYBR green was detected using the CFX96 Touch Real-Time PCR Detection System (Bio-Rad) and analyzed using CFX Maestro Software (Bio-Rad). The expression levels were quantified by ^ΔΔ^Ct method using *B2M* or *GAPDH* as reference gene.

### 4.7. Statistical Analysis

All values are presented as the mean ± SD. Comparisons between groups of pathological findings were analyzed using an unpaired two-tailed *t*-test (SPSS Advanced Statistics 17.0). A *p*-value of less than 0.05 was considered statistically significant.

## 5. Conclusions

Our study demonstrates the different phenotypes and pathological findings of DBP and IBP. In clinical practice, DBP predominantly presented as non-inflammatory BP. We demonstrated that vildagliptin decreased the expression of *LAMA3*, *LAMB3*, *LAMC2*, *DST*, and *COL17A* and proposed a new mechanism other than the inflammatory pathway in the pathogenesis of DBP. Furthermore, vildagliptin may also drive the IL-6-mediated inflammatory pathway toward DBP development. These findings explained how bulla formation in the presence of a few eosinophils infiltrated DBP lesions.

## Figures and Tables

**Figure 1 ijms-23-14101-f001:**
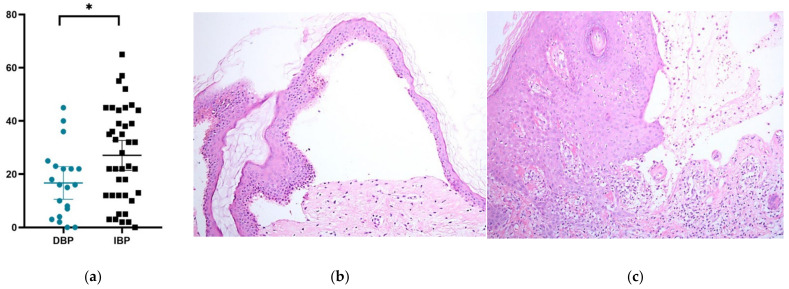
Histopathological features of DBP and IBP. (**a**) Scatter plot of infiltrating eosinophils under a high-power field (hematoxylin–eosin stain, 400×), (**b**) subepidermal blister with sparse eosinophil infiltration present in DBP (hematoxylin–eosin stain, original magnification ×100), and (**c**) subepidermal blister with high eosinophil infiltration in IBP (hematoxylin–eosin stain, original magnification ×200). * [*p* < 0.05].

**Figure 2 ijms-23-14101-f002:**
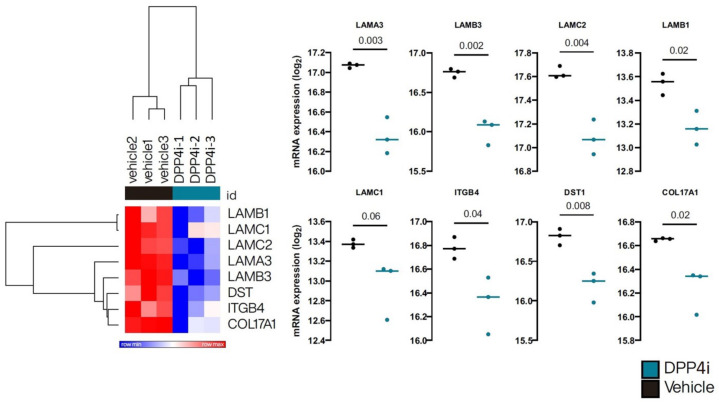
DPP4i treatment downregulates the expression of genes related to the dermoepidermal junction. Heatmaps showing the relative expression of *LAMB1*, *LAMC1*, *LAMC2*, *LAMA3*, *LAMB3*, *DST*, *ITGB4*, and *COL17A* with or without DPP4i treatment using Euclidean distances and hierarchical grouping. Scatter plots show the expression of individual genes with or without DPP4i treatment. Comparisons between groups were performed using an unpaired two-tailed *t*-test. *p*-values are marked on the graphs, and *p* < 0.05 was considered statistically significant.

**Figure 3 ijms-23-14101-f003:**
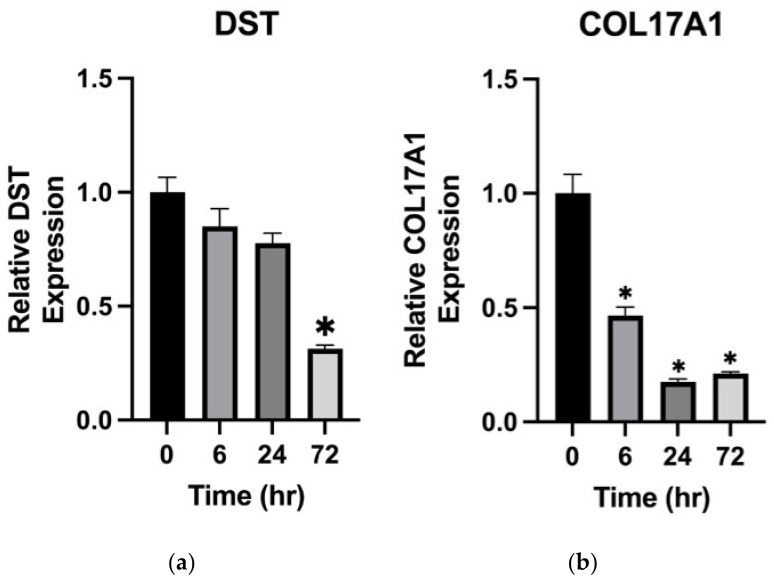
Vildagliptin decreased the expression of *DYS* and *Collagen 17A1* in a time-dependent manner. HaCaT was treated with Vildagliptin (5 µM) for 0, 6, 24, and 72 h. The expression of (**a**) *DYS* and (**b**) *Collagen 17A1* was checked by qPCR. * [*p* < 0.05].

**Figure 4 ijms-23-14101-f004:**
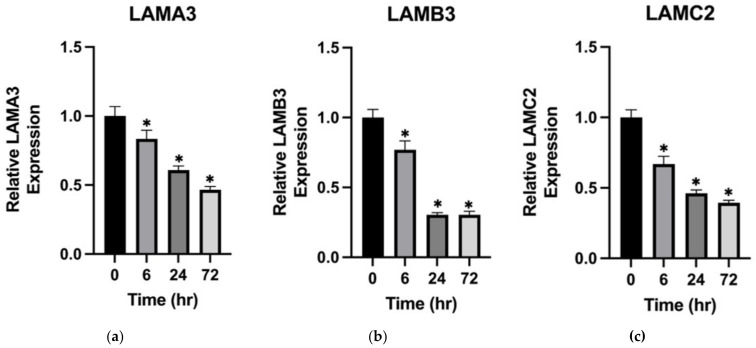
Vildagliptin decreased the expression of *LAMA3*, *LAMB3*, and *LAMC2* in a time-dependent manner. HaCaT was treated with Vildagliptin (5 µM) for 0, 6, 24, and 72 h. The expression of (**a**) *LAMA3*, (**b**) *LAMB3*, and (c) *LAMC2* was checked by qPCR. * [*p* < 0.05].

**Figure 5 ijms-23-14101-f005:**
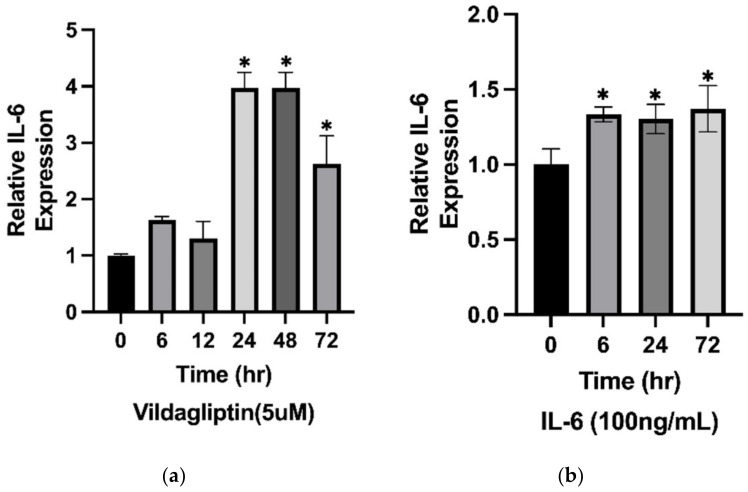
The expression of *IL-6* was stimulated in keratinocytes treated by either vildagliptin or IL-6. (**a**) Checked by qPCR, the expression of *IL-6* was found to have increased when HaCaT cells were treated by vildagliptin (5 µM). (**b**) By adding IL-6 (100 ng/mL) in the HaCaT cell culture medium, the endogenous *IL-6* increased in a time-dependent manner checked by qPCR with GAPDH as reference gene. * [*p* < 0.05].

**Table 1 ijms-23-14101-t001:** Comparisons between DBP and IBP on demographic data, clinical features, and laboratory data.

Characteristics	DBP	IBP	*p*
Total	20	40	-
Age at diagnosis, mean (SD), years	79.25(10.9)	79.93 (10.6)	0.8187
Male, n (%)	13 (65)	20 (50)	0.2787
Pruritus, n (%)	19 (95)	40 (100)	0.1591
Mucosa involved, n (%)	2 (10)	4 (10)	-
BPDAI scores, mean (range)			
Overall	37.1 (5–105)	36.33 (5–53)	0.8927
Erosions and blisters	24.6 (3–65)	16.68 (3–42)	0.0189
Urticaria and erythema	12 (0–40)	19.05 (2–46)	0.0183
Mucosal lesions	0.5 (0–5)	0.6 (0–7)	0.8359
Hematological involvements, n (%)			
Anemia	15 (75)	21 (52.5)	0.0966
Thrombocytopenia	1 (5)	5 (12.5)	0.3698
AEC, median (range), cells/μl	780 (0–3138)	470 (0–1672)	0.4216
AEC ≥ 500 cells/μL, n (%)	7 (35)	17 (42.5)	0.5837
AEC ≥ 1500 cells/μL, n (%)	2 (10)	1 (2.5)	0.2156

AEC: absolute eosinophil count; BP: bullous pemphigoid; DBP: dipeptidyl peptidase-4 inhibitor induced bullous pemphigoid; IBP: idiopathic bullous pemphigoid; BPDAI: Bullous Pemphigoid Disease Area Index.

## Data Availability

The data presented are available and can be provided by the corresponding author on request.

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
