# Peer review of "Dipeptidyl Peptidase-4 Inhibitor-Related Bullous Pemphigoid: Clinical, Laboratory, and Histological Features, and Possible Pathogenesis"

_ijms, 2022, doi:10.3390/ijms232214101_

Round 1

Reviewer 1 Report

There is a doubt in the information provided en la legend or caption of the photomicrographs of the histological sections and, it is respect to the magnifications of Image (c) -wich says it es 100x, when it is observed that is greater than the image of (a). Or, specify in each one the magnification.

Interesting analysis carried out on the characteristics of the BP, which allowed us differentiate the aspects of the origin of the pathogenesis.

Author Response

Thanks for your kind comment and reminder. We revised and rephrased our article accordingly. Figure 1. Histopathological features of DBP and IBP. (a) Scatter plot of infiltrating eosinophils under a high-power field (hematoxylin-eosin stain, 400X), (b) subepidermal blister with sparse eosinophil infiltration present in DBP (hematoxylin-eosin stain, original magnification ×100), and (c) subepidermal blister with much eosinophil infiltration in IBP (hematoxylin-eosin stain, original magnification ×200).

Author Response

  1. The sentence from abstract “Dipeptidyl peptidase-4 inhibitors (DPP4is) are widely-used antidiabetic agents” should be “widely used” without the dash.

<Response> Thanks for your kind comment and reminder. We revised and rephrased our article accordingly.

  1. A discrepancy of description of “DDP4is, DPP4i” from the first sentence. Please double check that and make it consistency.

<Response> Thanks for your kind comment and reminder. We revised and rephrased our article accordingly.

  1. “There was no significant difference in mucosal lesions between DBP and IBP” Why there is no difference of symptoms between the using of DBP and IBP for certain disease (mucosal lesions) and why there are big difference for others like “erosions and blisters”? (Line 77-83)

<Response> Thanks for your kind comment. We observed this interesting finding in our current the limitation of the number of patients. Hence, we did not observe the differences in other clinical presentations in these patients.  However, it should be interesting issue to address for future more recruiting patients.

  1. The author did not make a clear conclusion about their studies. Like the statement of “Furthermore, we also set up experiments to explore how DPP4is may affect human keratinocytes in the gene expression of relative factors for BP development, including cytokines and adhesion molecules distributed in the dermo-epidermal junction where the blistering of BP occurred.” In the final sentence of the introduction. The authors need to clearly state their hypothesis and possible experimental or statistical evidence, then to what extent, supporting their conclusion.

<Response> Thanks for your kind comment and reminder. We rephrased our introduction to make it clearer (lines 67-73).

  1. In line 50, the authors stated, “Similar results were also observed in Japan and China [18,19].” However, a further analysis should be provided to what extent of similarities between the studies (like number of patients, symptoms) to support the idea that “patients taking DDP4is had a higher risk of BP”

<Response> Thanks for your kind comment and reminder. We revised and rephrased our article accordingly as follows, “Similar results were also observed in Japan’s nationwide retrospective observational study [243 with DBP and 461 with IBP, vildagliptin (37.2%)] and China’s retrospective analysis of databases [114 with DBP, vildagliptin (52.63%)] [18,19].”

  1. The authors used “60 patients with BP, including 20 patients with DBP and 40 patients” was that number adequate to support their studies and conclusions? Why or why not. Would more data provide a more reliable conclusion?

<Response> Thanks for your kind comment and reminder. Considering the uncommon DBP cases, we did our best to collect 20 patients with DBP in our hospital. In the future, the multicenter study or international study would be performed to provide a more reliable conclusion.

  1. There is an interesting possibility that “lower number of eosinophils infiltrating the skins of patients in DBP than in IBP” and “other non-inflammatory pathways” might be related to “decreased gene expression such as LAMA3, LAMB3, LAMC2, DST, and COL17A1”. Could the authors provide more analysis about the relationships between these three parts? <Response> Thanks for your kind comment and reminder. In this study, we proposed the noninflammatory pathway that DPP4i decreased the genes related to the dermo-epidermal junction and the inflammatory pathway that DPP4i increased IL-6 expression and IL-6 could amplify IL-6 expression itself. Both findings explained that bulla appeared under a few eosinophils infiltrated in DBP lesions. It should be a beginning step to link these three parts and more experiments are needed to perform to verify our findings.

  1. They did conduct an in-depth analysis and discussions about the possibilities and differences in phenotypes and pathological findings of DBP and IBP and derived diseases. However, the conclusion is too weak and need better paraphrase their discoveries and why.

<Response> Thanks for your kind comment and reminder. We rephrased our conclusion to make it clearer. (lines 384-390).

Reviewer 3 Report

In the manuscript entitled "Dipeptidyl Peptidase-4 Inhibitor Related Bullous Pemphigoid: Clinical, Laboratory, Histological Features, and the possible Pathogenesis", the Authors investigated the role of eosinophil infiltration in driving inflammatory cascades in Taiwanese patients affected by DBP. The topic is interesting and the experimental study well done. As already reported by the authors "This finding implies that ethnic discrepancy plays a role in the phenotype of DBP", it would be more interesting to better discuss about their findings and those obtained from other ethnic groups.

Author Response

Thanks for your kind comment and reminder. We revised our current discussion about their findings and those obtained from other ethnic groups (Lines 209-221). In the future, the multicenter study or international study would be performed to provide a more reliable conclusion and answer the question of ethnic discrepancy.
